# An ESP Approach to Teaching Nursing Note Writing to University Nursing Students

**Shiou-Mai Su [1], Yuan-Hsiung Tsai [2] and Hung-Cheng Tai [3,4,*]**

1   Department of Cosmetic Science, Linkou Campus, Chang Gung University of Science and Technology, Taoyuan City 33303, Taiwan; susansu@gw.cgust.edu.tw
2   Department of Radiology, Chiayi Chang Gung Memorial Hospital, Chiayi 61363, Taiwan; russell.tsai@gmail.com
3   Department of General Education, Chiayi Campus, Chang Gung University of Science and Technology, Chiayi 61363, Taiwan
4   Joint-Appointment Research Fellow, Chiayi Chang Gung Memorial Hospital, Chiayi 613016, Taiwan
*   Correspondence: alextai@gw.cgust.edu.tw

**Abstract:** For nursing students in the vocational education system in Taiwan, English-language writing skills, whether for general or specific purposes, have long been ignored, which may jeopardize their success in medical-oriented courses and their future careers. This study designed a nursing note-writing course (NNWC) for university nursing students and explored the teaching outcomes of its implementation. The three main objectives were to (a) examine the effectiveness of the NNWC in enhancing learners' competencies; (b) survey learners' satisfaction with the NNWC, and (c) investigate learners' perceptions of the NNWC. In this action research, 49 students practiced five writing tasks while guided with five teaching tools, namely an online writing platform, multiple revisions, peer-review activities, and direct and indirect teacher feedback, for a semester. External examiners included a language teacher and a nursing professional, and the data-collection instruments used included a writing competence scale and a course satisfaction questionnaire. The results showed that the learners' writing competence significantly improved after the NNWC. They also demonstrated a fair level of satisfaction toward the NNWC. A total of 90% of the learners preferred online writing compared to traditional handwriting. The learners also indicated a preference for feedback from the teacher rather than from peers, and they perceived vocabulary capability to be crucial. ESP/ENP teachers are advised to consider the implementation of the NNWC when designing syllabi.

**Keywords:** nursing note-writing course; nursing education; EFL; ESP; ENP





## 1. Introduction

In the field of English for specific purposes (ESP), workplace English for medical and nursing professionals is a crucial subject that deserves more attention from language researchers and educators [1]. Yet, within the healthcare industry itself, English for medical purposes (EMP) has traditionally been the primary focus of research. Meanwhile, English for nursing purposes (ENP) has recently been recognized as part of EMP [2]. Nevertheless, most discussions of ENP seem to focus on nurses practicing in ESL (English as a second language) contexts in which most people are native speakers of English [3], whereas little literature can be found regarding nurses living in countries where English is spoken as a foreign language (EFL) [3].

Regarding the teaching and learning of ESP language skills in the EFL context of Taiwan, the previous literature on EMP has indicated that emphases are placed on the reading and writing skills required to meet physicians' needs [4,5]. On the other hand, ENP has typically focused on the listening and speaking skills needed to communicate with foreign patients and their families in clinical settings [1,6]. Despite these focuses, the

accurate understanding and use of English in written documents, such as doctor's orders, admission notes, patient histories, and so forth, are still important for nurses [6].

However, English writing skills training for nurses, including nursing note writing, has long been ignored in the research context [7]. Some may argue that local nurses do not need such writing skills, as they are not competent enough to handle this difficult productive skill. In addition, nursing school curricula rarely include writing skills training due to the scarcity of educational resources. However, considering globalization and increased needs for nursing professionals in many countries, addressing ENP-related needs has become increasingly important. For EFL nurses who would like to go abroad for nursing work, ENP can be an indispensable skill. Relatedly, nursing note-writing training seems to be a niche worth exploring to compensate for the current lack of ENP knowledge among many nurses.

Therefore, this study utilized a design of action research under the theoretical framework of TPACK model (technological, pedagogical, and content knowledge) to investigate the teaching of ENP writing to EFL nursing students who were interested in learning nursing note writing. The main aim of the study was to design and implement a nursing note-writing training course (NNWC), and the three primary, related objectives were: (a) to examine the effectiveness of the NNWC in enhancing learners' writing competencies; (b) to survey the learners' satisfaction with the NNWC; and (c) to investigate the learners' perceptions of the NNWC.

## 2. Literature Review

### 2.1. Needs Analysis of Nursing Note Writing

The quality of nursing notes (or documentation) is a critical indicator of the quality of patient care [8,9]. Clinically speaking, nursing notes are the main source of information directly pertaining to patients and thus can have a substantial effect on the quality of nursing care. Relatedly, nursing notes generally must be completed and filed according to relatively high standards in order to ensure the quality and safety of medical services [8]. At present, medical institutions around the world are increasingly focusing on improving the quality of nursing notes. The quality of such documentation can be judged in terms of three main aspects: the content, the documenting process, and the format or structure [9]. The content focuses on the completeness and accuracy of authentic data in the clinical setting. The documenting process emphasizes the integrity of the patient's data and the regularity of the data in the patient's record. The format/structure pertains to the presentation of patient information, such as its legibility and integrity.

Nursing note writing has a unique form of grammar, which is different from that in general writing, as the purpose of such notes is to communicate with other medical professionals in the most efficient way. The typical form of such notes is similar to that of the "telegraphs" used in the maritime or navigation industries, and readability for the general public is not of primary concern [10]. Yu, Su, and Chen [10] proposed 8 grammatical features in nursing note writing, including omitting the subject, omitting the object, omitting the subject and verb, omitting articles, omitting the verb "be", writing in the passive voice, using abbreviations, and writing with one tense. In addition, at the diction level, some higher frequency words, such as patients (PTs), doctors (Dr.), and nurses (N), are frequently abbreviated or even replaced with ellipses.

The nursing note is an important medical record that helps to connect nurses working in different shifts. As each note is patient-care centered, the paragraph construction is typically based on the nursing care history of a specific case during a given work shift. The work shift is normally divided into day shift, evening shift, and midnight (or night) shift. The note may consist of a message indicating what has been done in this shift, and what needs to be carried out during the next shift. Thus, the effectiveness and efficiency of nursing note writing are crucial for the transition of nursing works. The conciseness and precision of the nursing notes expression become critical for clinical practice. Besides,

nursing notes can be kept as an official document for future reference in case any argument should occur among staff or patients [11].

### 2.2. English as Foreign Language (EFL) Writing Teaching and Learning

In the process of learning how to write, particularly at the revision stage, teachers' feedback and comments play a crucial role for student writers. Various categorizations of teachers' comments and feedback have been proposed, for example, stand for or against the teachers' comments, praise versus criticism; oral and written responses; end and side commentary; responses only to linguistic errors and/or to content; explicit versus implicit suggestions; sentence-level or ideas and organization; personalized versus group feedback; expert or non-expert readers' feedback; peer's versus teacher's comments; diagnostic chart employment; minimal marking strategy; and code correction system [12]. Although there are disagreements among these approaches, generally, the effect of a teacher's comments and feedback is context-specific and is subject to the learners, teacher, classroom setting, culture, objectives, goals, and so on [13].

Feedback and comments from a teacher focusing on grammatical errors are helpful for EFL learners' improvement in learning to write. Some writing scholars, such as Lee [14], have lodged the criticism that writing teachers put too much emphasis on their students' grammatical problems; nevertheless, expert writers' feedback and comments on student-writers' syntax errors are still welcomed by both learners and teachers [15]. Hamed Mahvelati [16] and Han and Hyland [17] argued that learners valued grammatical corrections and feedback more highly than feedback regarding the other dimensions of writing. Research conducted in EFL contexts in Asian countries, such as Taiwan [18], has also tended to conclude that linguistic knowledge, especially regarding grammatical rules and vocabulary usage, is the most problematic area for students' writing. The authors of such research have suggested, relatedly, that although learners with different levels of English proficiency might have different major difficulties, for novice learners, writing performance will be greatly constrained by their linguistic knowledge or lack thereof [18].

A combination of indirect and direct feedback is suggested for the remediation of writing errors/mistakes. Direct correction involves the underlining, highlighting, and explicit correction of incorrect word choices or grammatical errors by writing teachers, while indirect correction refers to situations when teachers provide more implicit hints, such as placing a question mark next to, inserting an arrow next to, or underlining words or phrases containing errors/mistakes without also providing specific corrections for them [19]. Responding to advocates of direct correction, some researchers have suggested that teachers should avoid providing the correct linguistic forms/ideas directly and immediately. Instead, student writers should be allowed to have more time and space to learn via critical thinking, reflection, and self-exploration [15].

The utilization of a coding system that gives the students indirect hints about their errors/mistakes has been recommended in more recent research [20]. Similar to the other arguments regarding two extremes of useful teaching techniques, both direct and indirect correction can be worthwhile in different times, settings, and teaching contexts [21]. Thus, writing teachers should attempt to find a suitable way to integrate these two strategies.

### 2.3. Peer Review Activities

Peer review, also termed peer feedback or peer response, has been discussed in the literature regarding English writing teaching and learning over the past two decades. The earlier interest in PR was in regard to its use in traditional classroom settings, and both college-level and secondary students' writing learning seemed to benefit from this technique [22]. A large number of empirical studies concluded that PR supports the enhancement of writing learners' capabilities and efficacy in terms of cognitive, affective, social, and linguistic perspectives [23]. Through the process of drafting, peer-reviewing, giving/receiving feedback, and revision, learners can simultaneously increase their awareness of their audiences, develop positive attitudes about learning writing skills, develop critical

thinking abilities, complement their second language acquisition, pay greater attention to their own writing, and even help improve their own speaking skills [23].

Following the rapid development of computer and information technology, especially the widespread use of the Internet, increasing numbers of studies have reported on the issue of PR facilitated by electronic communication [24]. This movement from traditional classrooms to virtual space seems to have brought more advantages for the development of PR [24]. Researchers have claimed that online courses can not only maintain the benefits of the traditional face-to-face mode of peer response but also provide additional benefits stemming from the use of computer technologies, such as allowing students to work at any time and in any location while maintaining clear records of all the comments and ideas exchanged [25].

Likewise, both teachers' comments and PR are feedback worth considering in terms of their effects on student writing. Some studies have shown that student-writers prefer their teachers' comments to responses from their peers, while a few others have suggested that PR is beneficial for the construction of the writing process [25]. Taking a middle path, EFL writing learners can properly take advantage of both forms of feedback to improve their writing skills, especially in different social and cultural teaching contexts [13]. Moreover, the facilitation provided by online learning environments may help enhance the feasibility and quality of teaching efforts by freeing students from some course and syllabus constraints.

*2.4. Online Writing Platform*

Using technology to assist writing skills teaching and learning has become a prominent issue in the field of foreign language acquisition. With the continuous development of information and communication technology (ICT), technology-mediated instruction (TMI), which supports composing processes, has been widely discussed in recent decades [26]. A number of findings stated that TMI can be more effective than traditional paper-pencil methods in improving students' writing performance [27] and enhancing simultaneous student engagement [28]. Early TMI research on writing mostly used personal computer's word processing software, such as Microsoft Word, to teach writing. Bangert-Drowns [29] conducted a meta-analysis of 32 studies with experimental and control group design. The results showed that the experimental group design using word-processing software was more effective in improving students' writing performance than the control one, which applied the traditional paper-pencil method. Another meta-analysis research analyzed 26 studies of K-12 writing courses in U.S. primary and secondary schools from 1992 to 2002. These studies revealed students who used computer equipment to learn writing not only had higher motivations to devote themselves to writing but also had better skills regarding the length of the article, the depth of the vocabulary, and the quality of expressions [30].

Corresponding to the claims of many educational researchers, TMI may enable the act of sharing and reading of learners' writing works, which in turn helps to achieve the process of socialization. Although students seem to be silent and independent in TMI context, the learning environment created by the ICT is very suitable for promoting collaborative learning and peers interaction [31]. However, students need to have persistent and frequent interaction with peers, teachers, readers, etc., through writing and feedback. The interactive process accelerates their development of communicative competence and the progress of socialization [32]. In 2005, a large-scale notebook writing project was conducted in Maine, USA, and more than 100,000 middle school students participated. The results showed that students who wrote with laptops were generally better writers than those who wrote with traditional pen and paper. The content and depth of writing were better in the laptop group, too. In other words, students who received TMI in learning writing may have more opportunities to go through the process of information sharing and social interaction and therefore better chances to grow into better writers [33].

### 3. Methods

#### 3.1. Research Design

This action research aims to develop and implement a nursing note-writing course (NNWC) applying the TPACK (Technological Pedagogical Content Knowledge) model as the research framework (see Figure 1). The action research method was one in which the researcher repetitively revised and improved the teaching actions through the spiral process of planning, implementation, observation, evaluation, and reflection [34]. Aside from the cycle of action research, the TPACK model [35] integrated various pieces of knowledge in the teaching and learning processes, including technology, pedagogy, content knowledge, technological pedagogy, pedagogical content, technological content knowledge, and technologically pedagogical content [7]. Technologically, the online writing platform was utilized. Pedagogically, the notion of "process writing approach" [7,23] was adopted. Contently, the EGP and ESP/ENP knowledge were both taught in the course. The effectiveness and efficiency of the application of this TPACK model have been well recognized in many educational studies [7,36,37].

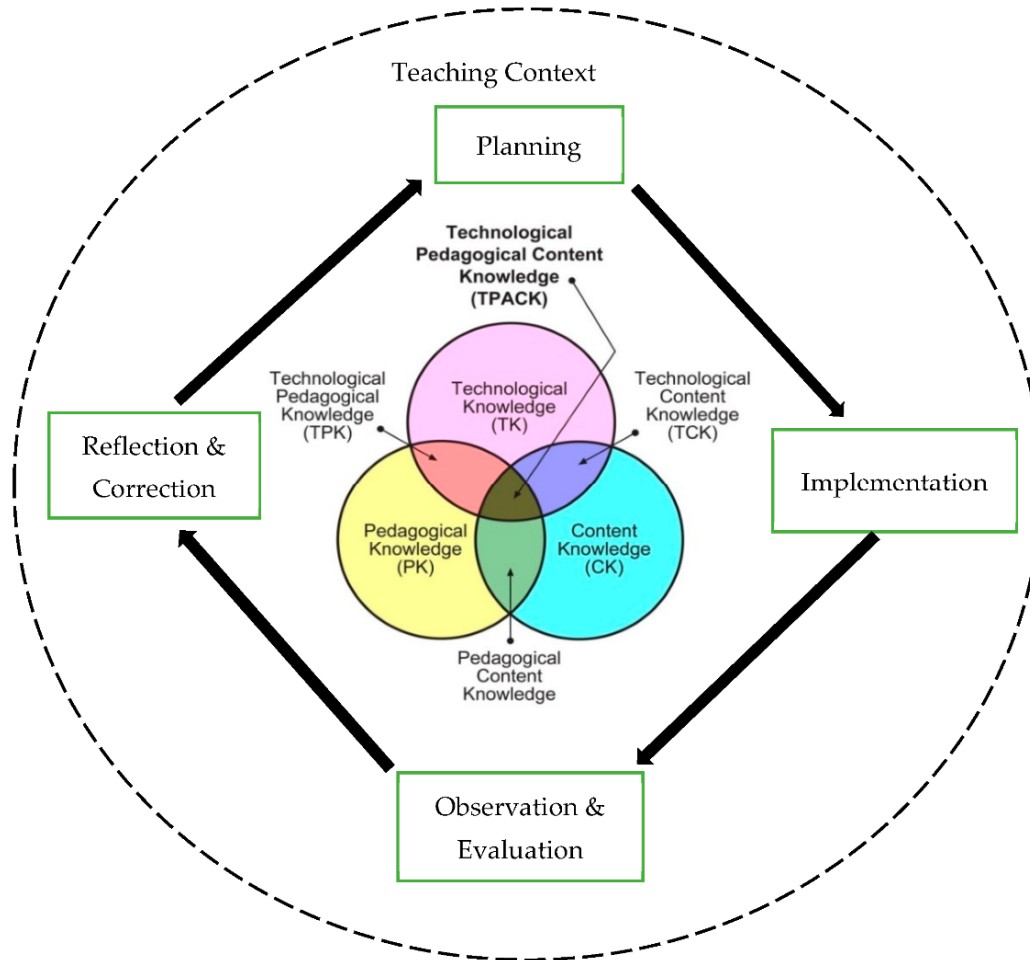

**Figure 1.** The research framework of this NNWC course.

A comprehensive instruction method integrated with various teaching strategies and writing tasks was designed for the special needs in the ESP/ENP context. The research procedure of this NNWC is shown in Figure 2.

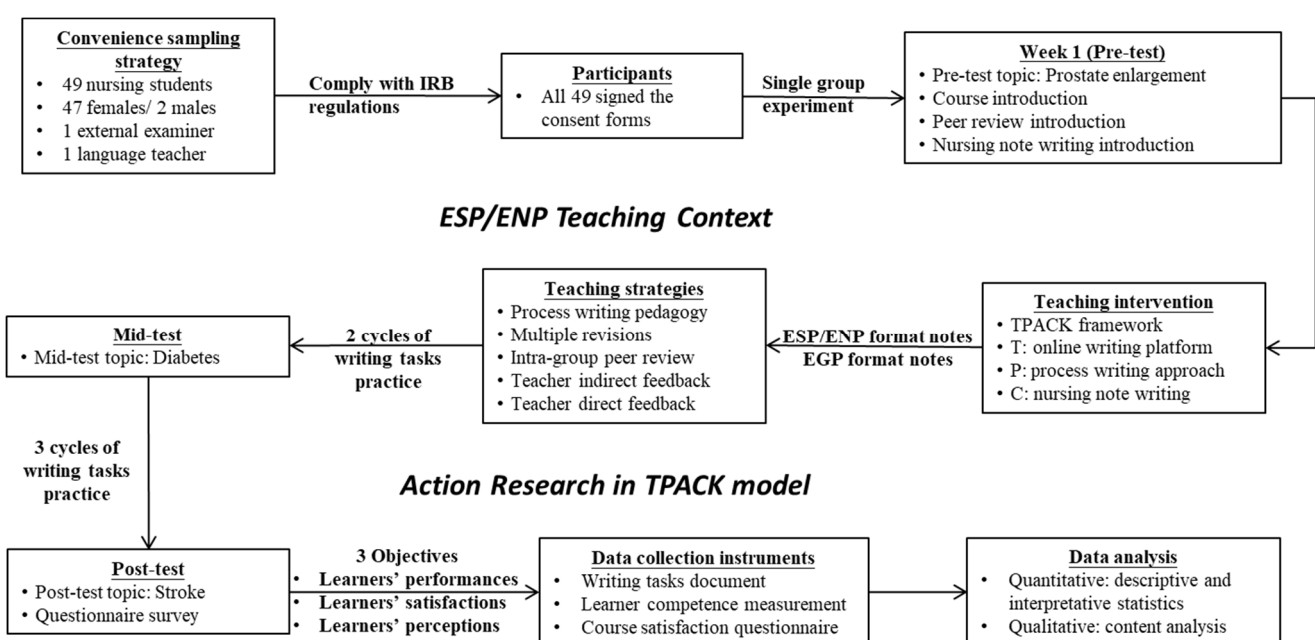

**Figure 2.** Research procedure of this NNWC course.

### 3.2. Participants

The participants consisted of 49 nursing majors. To recruit nursing students who were interested in learning nursing note writing, one classroom consisting of 49 students in the research context was conveniently sampled at the target university. As the regulations of the relevant IRB (Institutional Research Board) stipulate 20 years of age as the age of adulthood, we only invited 2-year vocational students who had already completed 5-year junior college programs to take part in the study. Most of the students (47 of the total of 49) were females, while all the participants were aged from 21 to 25 years old, with the majority being 21 years old.

Normally, nursing students attending science and technology universities in Taiwan have good levels of nursing professional knowledge but are less competent in English as a foreign language. They have studied nursing specialties intensely (for at least 20 h per week) for over 3 years and have obtained nursing licenses issued by the relevant Taiwanese governmental agencies. To pass the national licensure examinations written in Mandarin, nursing students need to be competent in nursing professional knowledge. However, most nurses are not competent enough in English to comprehend the English versions of these tests [5,6] although most have studied English as a foreign language for at least 8 years, including 3 years in junior high school and 3 years in a 5-year junior college program with 2 h lectures every week.

The nursing students' EGP proficiency was not satisfactory. The university graduation threshold of EGP was GEPT (General English Proficiency Test) elementary level, CEFR (Common European Framework of Reference for Languages) A2, or TOEIC (Test of English for International Communication) 235 with 115 in listening and 125 in reading minimally. Under this requirement, less than 20% of students could pass it. Most of them chose to take another ESP/ENP vocabulary test, named PVQC (Professional Vocabulary Quotient Credential), or a campus English test held by the university. Of these language examinations, writing skills is excluded, and nursing students are often beginners in English writing [7,13,18]. The nursing students' English test scores at the university are shown in Appendix F.

The students' nursing note tasks were evaluated by an external examiner, a nursing teacher who was educated in the United States. This nursing teacher is familiar with the conventions of nursing notes and could provide proper suggestions regarding the content of nursing notes. She could also read and judge the quality of nursing notes in English given

her training background and expertise. Her participation helped to ensure not only the reliability and validity of the writing tasks but also the research conducted for this project.

The language teacher who was also the action researcher has been teaching at the university in question for more than 20 years and is a full-time faculty member who is very familiar with the backgrounds, needs, and characteristics of the target learners. He played a crucial role in the process of providing writing teaching, learning, and feedback during the study and in conducting the associated research as well.

### 3.3. Teaching Interventions

### 3.3.1. The TPACK Framework

Technologically, the use of an online writing platform was implemented (see Figure 3). As the writing practice projects took a long amount of time to complete, the teaching and learning activities took place both inside and outside of the writing classrooms. To make it feasible for some parts of the course to be conducted outside the classrooms, an online writing platform was designed. We utilized existing software—e-campus software—provided by the university to set up the virtual space for the writing teaching and learning media [7]. Through this platform, the nursing students could log in to their own accounts, read handouts and other information released by the teacher, submit their writing tasks, give/receive feedback to/from their peers, and get feedback/comments from their teacher.

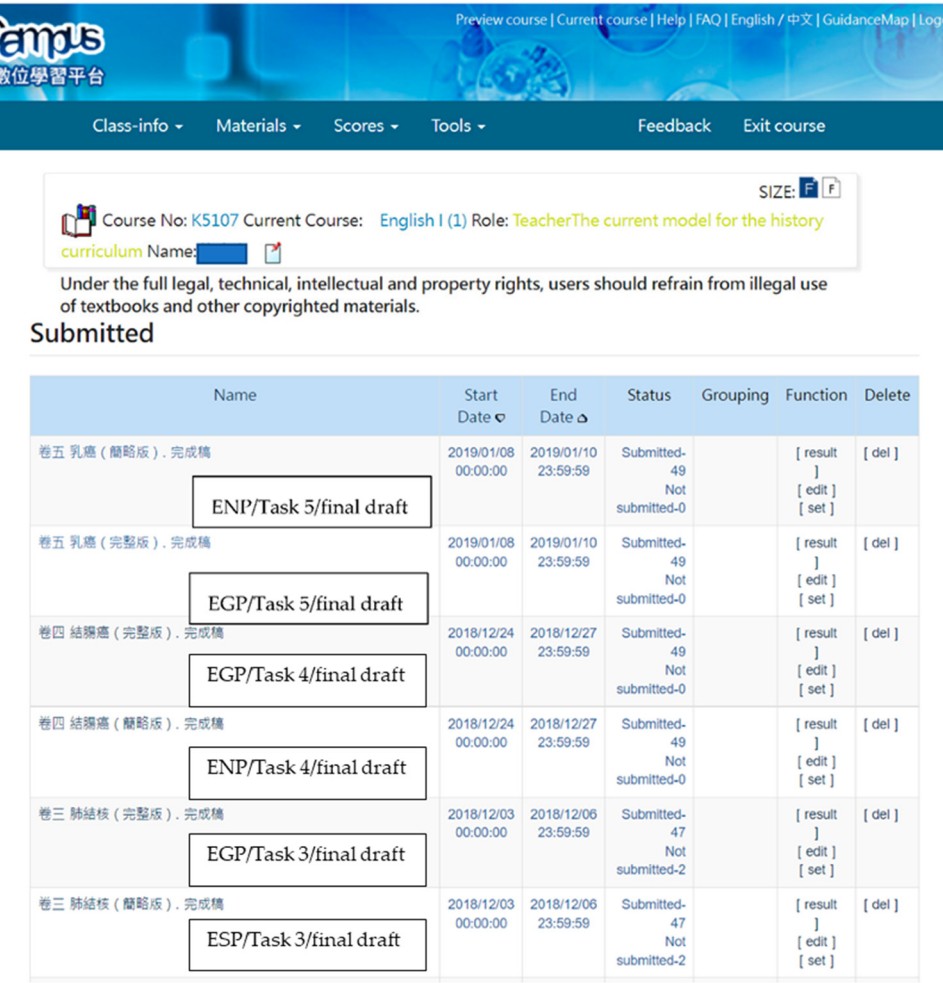

**Figure 3.** Online writing teaching and learning platform.

Pedagogically, the process writing approach was applied in the course. Process writing views the learning of writing as a process and progression rather than merely focusing on performance as suggested by the traditional product approach [38]. The psychological

hypothesis of this pedagogy is based on a complex model of the writing process, involving writers' cognitive processing, environmental influences, and affective factors [23]. To help the participating nursing students scaffold the learning of nursing note writing, four strategies were applied, namely the writing of multiple drafts, intra-group peer review, indirect teacher feedback, and direct teacher feedback described as follows.

### 3.3.2. Four Teaching Strategies

The first strategy consisted of the writing of multiple drafts for each writing task's teaching and learning. One potential feature of the process writing approach was the production of multiple drafts during the writing teaching and learning progression [39]. As writing can be seen as a process of thinking, reflection, and discovery, revisions of the drafts according to the feedback and comments inside and outside of the writers' cognitions were a must. The nursing students started by writing their first drafts and then submitting them to the online platform. The peers of each student then gave him/her first-time feedback to help improve the quality of the given first draft. Second drafts were then submitted to the platform after corrective revisions were made. The teachers' indirect feedback and comments were then used as a reference in making further corrections. Third drafts were then prepared based on the teachers' implicit feedback. The teacher then proofread these third drafts to identify previously unidentified errors and provide direct feedback accordingly (see Figure 4). These four rounds of revisions should have enabled the learners to obtain insights regarding their linguistic errors as well as nursing knowledge expressions.

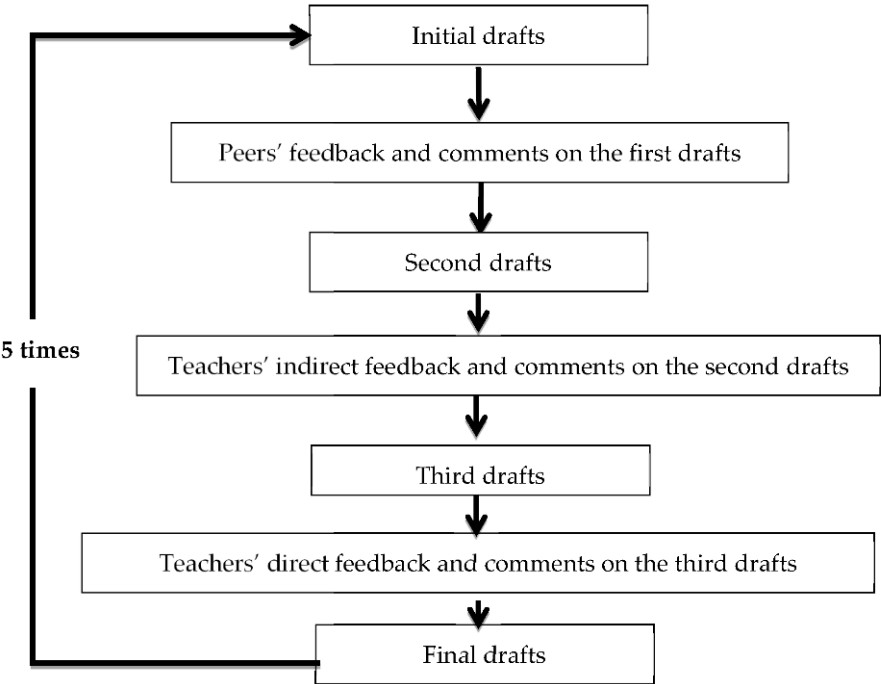

**Figure 4.** Multiple drafts teaching strategy.

The five repeated writing cycles correspond to the design of five writing tasks that are stated in the latter section of "nursing note-writing tasks". As an action research design, the five writing cycles also served another purpose of a spiral cycle of planning-implementation-observation-correction as shown in Figure 1. The focus of the feedback and comments became slightly different in each round as the nursing students were learning progressively.

The second strategy consisted of the use of intra-group peer review activities. Although the appropriate use of PR (peer review) in the field of EFL writing is debatable, a recent study showed that the integration of PR with teacher feedback (TF) yielded better results than the use of TF alone [13]. The value of PR seems to lie in providing more effort and study time for the learners to think and reflect. Therefore, we designed an intra-group

PR activity with three to four members for each group (see Appendix B). To ensure its success, PR training was supplied at the beginning of the course, which was intended to enable the nursing students to become familiar with the feedback forms and techniques (see Appendix B).

The third strategy consisted of providing indirect comments and feedback from the teacher. The value of and appropriate way to provide written corrective feedback (CF) have long been discussed in the field of English language teaching [12], with the importance of CF for the learning of writing being commonly acknowledged. When acting as mediators, models, experts, and facilitators, language teachers' linguistic knowledge is doubtless helpful for the stages of drafting and revising. Since we took on the notion of process writing emphasizing the feature of discovery in this study, it was thought that the teacher's indirect feedback could provide more time and space for the novice writers to solve the problems that the teacher identified. To achieve this, the teacher only marked "codes" on errors and mistakes instead of giving the "right answers" immediately. Such codes could consist of, for example, "TS" for a tense error, "SP" for a misspelling, "SV" for the incorrect use of a singular verb, and "PP" for the incorrect use of a preposition, etc. The students could then try to find out the correct answers from their peers, friends, the internet, or even the teacher.

The fourth strategy used consisted of the teacher providing direct feedback and comments. Following the stage of indirect feedback, which allowed the students to discover their own mistakes, the teacher performed final checks and provided CF explicitly. Ideally, the final drafts should have been readable without too many errors, and we expected better results after more of the writing tasks had already been completed. The final drafts were assessed by an external examiner to obtain a more objective judgment of the quality of the given task.

### 3.3.3. Nursing Note-Writing Tasks

Contently, there were five writing tasks that the students completed to practice writing—two took place before the midterm test, and the other three were performed after that (see Appendix A). The topics for these five nursing-note writing tasks covered five major scenarios of medicine [40]. All the drafts, including revisions, were kept and tracked on the writing platform for further analyses. Two formats of tasks serving different teaching purposes were given for each writing topic: one for EGP, which entailed writing in a formal and full writing style (see Appendix B), and the other for ESP/ENP, which entailed writing in an informal style with abbreviations (see Appendix C).

The topic of the first task dealt with the digestive system (which was covered in a gastrointestinal unit). The learners were taught about the conventions of nursing notes written regarding a scenario of an associated specialty. In the scenario, a patient with liver cirrhosis is visiting the gastroenterology department. Some gastrointestinal symptoms, such as passing black stool and vomiting blood clots, were described when admitting the patient to the hospital. Vital signs taking was then practiced, and the results were recorded accordingly. The same was done with physical examinations. After receiving the doctor's diagnosis and order, further treatments and caring procedures were carried out. Finally, discharge teaching was provided once the patient was qualified to be discharged. Such a typical cycle of nursing care routines was integrated into the writing activity.

Consequently, the second topic was related to the heart and circulation systems (which was covered in a cardiovascular unit). In this unit, a patient with chest pain, cold sweats, difficulty breathing, and feeling faint at work was the focus. Diagnostic tests demonstrated some more causes of associated disorders in depth. The administration and teaching of medications for different types of cardiovascular patients were demonstrated and performed. The students also learned how to take nursing notes for these procedures. The third topic was related to a respiratory unit concerning the lungs and breathing system. For this scenario, a male patient had been coughing for half a year and had coughed up dark red sputum occasionally. He was diagnosed with tuberculosis (TB) and was cured

in a respiratory ward with special equipment. For public safety reasons, he was traced for at least 6 months after being discharged. Note-taking regarding all the symptoms, medications, treatments, and healthcare teaching was included in the task. The fourth topic was related to a colon and rectal surgery unit dealing with a patient suffering from colon cancer, and the fifth was related to a breast surgery unit about treating a female client who was experiencing symptoms of breast cancer.

All eight tasks covering various topics had been carefully designed, tested, and widely adopted and taught in the teaching context in Taiwan. As they have been part of an ESP textbook published for over 10 years, such a rigorous development process had been experienced, including expert consultation, large-scale implementation, and continuous revisions, that the validity and reliability of the tasks should have been achieved to some extent.

*3.4. Data Collection and Processing*

3.4.1. Learners' Competencies

To assess the progress of the various learners' learning, three nursing note-writing examinations were conducted as pre-, mid-, and post-tests (see Appendix A). To comply with the process writing approach, one more mid-term test was also set up as a form of measurement feedback along with the commonly applied pre- and post-tests. The pre-test topic was a patient with prostate problems, which provided a medical context for a urological unit [40]. In the middle of the course, the nursing students were tested on the topic of a patient suffering from diabetes mellitus, a scenario from an endocrine unit. At the end of the training sessions, they were tested on the topic of a patient with a stroke, a scenario from a cardiovascular unit [40].

A set of competence criteria specified for the nursing note writing was designed by the researchers. While the conventions of nationwide EFL writing-test criteria, which contain the six dimensions of "content", "structure", "grammar", "diction", "mechanics", and "holistic" [41], were used in designing this set, a few aspects of those criteria were subsequently modified. As the medical and nursing context had been fully given and the writing purpose was to communicate with the nursing professionals, the "structure" dimension dealing with the structure of topic/concluding/developmental sentences was not particularly crucial anymore. On the other hand, the ESP/ENP grammar used was quite different from that of EGP, as nursing note writing omits subjects/objects/articles for the sake of conciseness and briefness. Additionally, abbreviations of regular words and terms are also frequently shown on nursing notes such that the "diction" dimension needed to be slightly modified as well, while the "mechanics" dimension dealt with the punctuation, capital letters, transitional words, and discourse markers. The peer-review form showed detailed items of the assessment criteria (Appendix D). As indicated in that form, the evaluation criteria covered five dimensions—"content", "grammar", "diction", "mechanics", and "holistic"—based on five scales of competence from 0 (lowest) to 5 (highest).

3.4.2. Learners' Perspectives

A course satisfaction questionnaire (CSQ) was designed by the researchers to investigate the learners' perceptions of the teaching course. The CSQ consisted of two sections: one a quantitative survey containing 26 items to assess the learners' satisfaction levels and the other a qualitative survey consisting of 6 open-ended questions (see Appendix E).

To increase the validity and reliability, three experts, including a nursing professor, a medical doctor, and a linguist, were involved to provide suggestions for the CSQ. A pilot survey with 15 students before the commencement of the study revealed satisfactory reliability of Cronbach's alpha value as 0.94. The original version of the quantitative section contained 30 items, which were reduced to 26 after the analysis of EFA (explorative factor analysis) of this CSQ.

### 3.4.3. Data Processing

Quantitative statistical analyses, including descriptive and interpretive statistics, were performed using SPSS version 21. A repeated-measure MANOVA (Multivariate Analysis of Variance) was opted for since the five measurements of the learners' writing competencies were included as dependent variables. Then, post hoc analyses were performed to further compare the effects in every competence dimension (i.e., content, grammar, diction, mechanics, and holistic). The significance level was set at 5% ($p = 0.05$).

Qualitatively, the CSQ was analyzed by means of content analyses [42]. The six steps involved were: (a) each original record was read carefully and repeatedly to comprehend each learner's perceptions; (b) specific comments and repeated descriptions were marked; (c) patterns of perceptions were coded to form a subtheme; (d) relationships between each subtheme were examined carefully; (e) higher-level themes were established based on the context of the subthemes; and (f) the themes of learners' perceptions were reviewed again.

### 3.5. Ethical Considerations

Before the commencement of the study, Institutional Research Board (IRB) permission granted by Chang-Gung Memorial Hospital was obtained to conduct the study ethically. All the study procedures complied with the IRB regulations. The main goals, implementation procedures, and possible risks of the project were explained to the participants. They had to sign the consent forms embedded in the questionnaire, which informed them of their freedom to decide whether to participate in the study or not. They understood their right to withdraw from the study at any time and to have their data withdrawn even after the study had been completed. They were further informed that the collected data would be used solely for academic purposes and kept confidential without publicly identifiable information unless their authorization to do otherwise was obtained.

### 4. Results and Discussion

#### 4.1. Learners' Writing Competencies

The learners' competencies about both EGP and ESP/ENP nursing note writing exhibited significant improvement after they had completed the semester long NNWC. The results of the repeated-measures multivariate analysis of variance (MANOVA) shown in Table 1 demonstrated they had achieved statistically significant improvement in both EGP and ESP/ENP nursing note writing ($p < 0.001$). This suggests that the learners' writing competencies had been enhanced for at least one among the three repeated measurements. The results thus suggest that the NNWC was an effective approach for helping improve the nursing students' writing competencies although part of the course took place outside the classroom through the implementation of an online teaching and learning platform provided by the nursing university, which served as the research context. The current finding corresponds to the previous research regarding TMI and ICT [26,31,32].

**Table 1.** Three times repeated-measure MANOVA (pre-, middle-, and post-test).

| Format | Effects | Test Method | Value | F Value | df | $p$ | Partial $\eta^2$ | Observed Power |
|--------|---------|-------------|-------|---------|-----|------|------------------|----------------|
| EGP | Within Subjects | Wilks' Lambda | 0.042 | 166.503 | 6 | 0.000 ** | 0.958 | 1.000 |
| | Between Subjects | Wilks' Lambda | 0.130 | 23.776 | 11 | 0.000 ** | 0.870 | 1.000 |
| ESP/ENP | Within Subjects | Wilks' Lambda | 0.002 | 1250.848 | 5 | 0.000 ** | 0.998 | 1.000 |
| | Between Subjects | Wilks' Lambda | 0.003 | 5176.303 | 10 | 0.000 ** | 0.997 | 1.000 |

Note: ** $p < 0.001$.

Judging from the F values obtained, the students achieved much higher scores for the ESP/ENP format than for the EGP format, indicating differing effects for the two formats. Specifically, the learners had more significant progress in learning ESP/ENP than in learning EGP.

To further investigate which part of the NNWC might have best helped the learners to boost their performances, the descriptive statistics might supply another form of evidence. Table 2 shows that the learners' writing performances for both the EGP and ESP/ENP formats exhibited significant improvements between the pre-test and mid-test in every dimension; however, their scores were maintained on a similar level between mid-test and post-test. At the beginning of the course, the learners seemed not yet ready to combine their previous knowledge about the nursing profession—or the schemata—with the English language [8]. The schemata knowledge may contain, for example, clinical scenarios, caring procedures, medical terminologies, patient communications, etc. [5]. The pre-test performance was miserable in general, as most of the nursing students could not write sufficiently well to meet the criteria designed for either format. The scores were poor for the ESP/ENP format, with the pre-test mean score being 0.22 (holistic), which implied that the nursing teacher gave the nursing students extremely low marks ranging from 0 to 0.6. On the other hand, for the EGP format, the pre-test mean was 1.69, and the highest scores were as high as 3.5. Regardless of the potential bias perceived by the assessors, the nursing students had some thoughts about the EGP already but no idea at all in terms of the ESP/ENP. As such, most of the learners left their papers empty for the ESP/ENP format or wrote the same content for both formats.

**Table 2.** Descriptive statistics of the pre-test, middle-test, and post-tests.

| Format | Test Order | Dimension | N | Min | Max | Mean | SD | Format | Test Order | Dimension | N | Min | Max | Mean | SD |
|---|---|---|---|---|---|---|---|---|---|---|---|---|---|---|---|
| EGP | Pre-test | Content | 49 | 0.0 | 4.5 | 2.02 | 1.26 | ESP/ENP | Pre-test | Content | 49 | 0.0 | 0.8 | 0.27 | 0.29 |
| | | Structure | 49 | 0.0 | 4.5 | 2.09 | 1.28 | | | Structure | 49 | 0.0 | 0.6 | 0.20 | 0.21 |
| | | Grammar | 49 | 0.0 | 3.2 | 1.51 | 0.95 | | | Grammar | 49 | 0.0 | 0.8 | 0.26 | 0.26 |
| | | Diction | 49 | 0.0 | 3.5 | 1.61 | 0.98 | | | Diction | 49 | 0.0 | 0.6 | 0.18 | 0.19 |
| | | Mechanics | 49 | 0.0 | 1.8 | 1.22 | 0.54 | | | Mechanics | 49 | 0.0 | 0.6 | 0.18 | 0.20 |
| | | Holistic | 49 | 0.0 | 3.5 | 1.69 | 0.97 | | | Holistic | 49 | 0.0 | 0.6 | 0.22 | 0.22 |
| | Mid-test | Content | 49 | 1.6 | 4.5 | 3.26 | 0.58 | | Mid-test | Content | 49 | 3.0 | 4.5 | 3.61 | 0.37 |
| | | Structure | 49 | 1.6 | 4.2 | 3.27 | 0.55 | | | Structure | 49 | 3.0 | 4.4 | 3.48 | 0.42 |
| | | Grammar | 49 | 1.2 | 3.2 | 2.14 | 0.41 | | | Grammar | 49 | 4.2 | 5.0 | 4.78 | 0.29 |
| | | Diction | 49 | 1.2 | 3.6 | 2.18 | 0.50 | | | Diction | 49 | 3.0 | 4.2 | 3.67 | 0.31 |
| | | Mechanics | 49 | 1.0 | 2.0 | 1.81 | 0.25 | | | Mechanics | 49 | 3.0 | 4.0 | 3.40 | 0.30 |
| | | Holistic | 49 | 1.3 | 3.4 | 2.53 | 0.39 | | | Holistic | 49 | 3.3 | 4.2 | 3.79 | 0.25 |
| | Post-test | Content | 49 | 1.0 | 5.0 | 3.53 | 0.95 | | Post-test | Content | 49 | 3.0 | 4.8 | 3.83 | 0.39 |
| | | Structure | 49 | 1.0 | 5.0 | 3.06 | 1.00 | | | Structure | 49 | 2.2 | 4.4 | 3.51 | 0.45 |
| | | Grammar | 49 | 1.0 | 5.0 | 2.51 | 1.11 | | | Grammar | 49 | 4.2 | 5.0 | 4.82 | 0.21 |
| | | Diction | 49 | 1.0 | 4.0 | 1.92 | 0.83 | | | Diction | 49 | 2.0 | 4.8 | 3.67 | 0.49 |
| | | Mechanics | 49 | 1.0 | 5.0 | 2.80 | 1.16 | | | Mechanics | 49 | 2.0 | 4.6 | 3.54 | 0.59 |
| | | Holistic | 49 | 1.0 | 5.0 | 2.88 | 0.90 | | | Holistic | 49 | 2.8 | 4.7 | 3.88 | 0.35 |

Nevertheless, following the NNWC instruction, the learners exhibited significant progress, from turning in almost-blank papers to being novice writers after merely two practice tasks. Their mean EGP score increased to 2.53, and their mean ESP/ENP score reached 3.79 (holistic) for the mid-test. Obviously, the learners' performance in the ESP/ENP format had tremendously advanced to fairly high levels based on their new knowledge of how nursing-note grammar rules are combined with medical terminologies and abbreviations [10]. One possible cause for the relatively high jump in the ESP/ENP scores might be that the ESP/ENP writing required different knowledge (i.e., the schemata) from that required for the EGP writing [1]. Moreover, the learners' weaknesses in EGP vocabulary and EGP grammar were not critical hindrances in performing the ESP/ENP tasks [1]. Once the nursing students understood the rules and requirements of the ESP/ENP format, the "underestimate" of their competencies at the beginning was rapidly improved upon [10]. On the other hand, enhancement was also observed for the EGP format despite the lack of EGP linguistic capabilities having potentially curtailed the learners' progress to a certain extent compared with the ESP/ENP format [18].

Furthermore, from the mid-test to post-test, the learners' performances remained on roughly the same level without showing statistically significant progress. Despite this, the nursing students had still gained slight improvements in both the EGP (holistic = 2.88) and

ESP/ENP (holistic = 3.88) formats at the end of the NNWC. As this course was mainly aimed at enhancing the nursing students' writing competencies regarding nursing notes, the post-test outcomes were gratifying to the extent that they improved from nearly zero to grades of 4 out of 5 for the ESP/ENP format. The teaching goal was thus partially satisfied. For EGP performance, on the other hand, EFL vocabulary and grammar competencies were not easy to develop in a short period [7].

Another possible reason for the insignificant improvement of the post-test, according to the researchers' observations, could be related to the learners' characteristics. Thanks to the development of information technology, which speeds up everything, including learning [34], our students are becoming fast learners in some ways but are also less patient about completing learning drills. They thus tended to perform problem-solving more effectively than engaging in traditional forms of practice [32]. As such, once they believed they had comprehended the major features of writing a nursing note, they could lose interest and focus.

Additionally, the EGP test results show that the mean scores for "diction" and "structure" decreased in the post-test compared to the mid-test although they were still higher than the pre-test. Regarding the diction dimension, the nursing students' EGP vocabulary competencies were not sufficient in general [5]. At the beginning of the EGP writings, they tended to use simple words so that they did not make many errors and mistakes, and they might focus more on the words' choices and spelling. However, after three more rounds of practice, they became more familiar with the writing formats so that they might start to use some relatively difficult words to try to correspond to their cognitive thoughts. Under this circumstance, they might make more mistakes.

In terms of the "structure" dimension, the draft of topic/developmental/concluding sentences was the focus. Since their native language, Mandarin, does not have this kind of structure, students found it easy to forget this dimension. Without repeated reminders of this dimension, they showed a decrease at the post-test. Besides, students had to pay more attention to the other four dimensions in which we could observe improvements. The two dimensions of "diction" and "structure" were relatively ignored.

*4.2. Learners' Satisfaction Levels with the Course*

The participating nursing students indicated a moderate level of satisfaction (mean = 3.79) with the NNWC as measured by the CSQ, and the questionnaire had good reliability (Cronbach's α = 0.94). The result is consistent with some previous studies in a similar research context [7,13]. Table 3 ranks the 26 items of the CSQ sorted by mean scores. The learners tended to agree that they learned more from the language teacher and the TAs than from their peers, which has been well discussed in the literature [16]. The teacher/TAs' efforts and guidance were generally recognized by the learners, and most of the positive attitudes stemmed from this (e.g., item 20 = 4.27; item 22 = 4.24). In contrast, the learners did not like the feedback from their peers such that most of the lower scores were generated from the associated questions (e.g., item 2 = 3.06; item 7 = 3.24). The hesitation toward peer review/feedback has been repeatedly reported by the previous research [25]. Meanwhile, regarding the syllabus and course design, the learners had relatively neutral but still positive levels of satisfaction (e.g., item 15 = 3.78; item 14 = 3.73).

*4.3. Learners' Perceptions toward the Writing Course*

After a qualitative analysis of the six open-ended questions embedded in the CSQ was performed, the results were briefly discussed. The first question asked the participants about their preference between online and traditional writing. In all, 90% of the students responded that they preferred writing online to traditional paper-pencil writing. Their responses, for example, included:

> "I can open the files and revise the drafts directly on the computer without printing anything out. This saves money and is also environmentally friendly." (ID. 13).

"When I need a word to express my thought, I can look it up in the online dictionary immediately. Online writing can help me increase the amount of vocabulary effectively." (ID. 7).

"There are no space and/or time restrictions to do online writing, and the tasks can be revised/proofread/marked efficiently." (ID. 20).

**Table 3.** Descriptive statistics of the course satisfaction questionnaire.

| Mean | Lowest | Highest | Range | Low/High | Variance | Items | N | Cronbach's $\alpha$ | |
|------|--------|---------|-------|----------|----------|-------|---|---------------------|--|
| 3.79 | 3.06 | 4.27 | 1.20 | 1.39 | 0.108 | 26 | 49 | 0.94 | |
| Item | | | Questions | | | | | Mean | S.D. |
| 20 | I feel that I have learned more from teachers' feedback and TAs than I from my classmates. | | | | | | | 4.27 | 0.75 |
| 22 | When I am correcting my writing, I will take into account the opinions of teachers and TAs. | | | | | | | 4.24 | 0.62 |
| 11 | In the process of writing, I feel that I need to strengthen my grammar and vocabulary. | | | | | | | 4.22 | 0.62 |
| 8 | The feedback from the TAs is very helpful for my writing improvement. | | | | | | | 4.16 | 0.62 |
| 12 | Since this semester, I have been very serious about writing activities. | | | | | | | 4.14 | 0.64 |
| 25 | The amount of TA feedback is very appropriate. | | | | | | | 4.08 | 0.78 |
| 21 | In general, the implementation of teacher and TA feedback activities are very helpful for our writing after the English class. | | | | | | | 4.02 | 0.71 |
| 13 | Since this semester, classmates have been very serious about writing activities. | | | | | | | 4.00 | 0.64 |
| 23 | The design of TA feedback is very appropriate. | | | | | | | 3.98 | 0.80 |
| 26 | The amount of teacher feedback is very appropriate. | | | | | | | 3.94 | 0.82 |
| 9 | The feedback from the teacher is very helpful for my writing improvement. | | | | | | | 3.92 | 0.70 |
| 24 | The guidance and response of the teacher's feedback are very appropriate. | | | | | | | 3.90 | 0.79 |
| 6 | I believe it is important to learn how to accept writing feedback from others. | | | | | | | 3.88 | 0.63 |
| 15 | In general, the implementation of writing activities helps me improve my English writing skills. | | | | | | | 3.78 | 0.62 |
| 14 | In general, the implementation of writing activities is very helpful for my English improvement. | | | | | | | 3.73 | 0.63 |
| 3 | I really like to receive writing feedback and opinions from the TAs. | | | | | | | 3.71 | 0.76 |
| 5 | I believe that learning how to give feedback to others' writing is very important. | | | | | | | 3.69 | 0.79 |
| 4 | I really like to receive feedback and comments from the teacher. | | | | | | | 3.61 | 0.72 |
| 16 | In general, the implementation of writing activities is very helpful for my future clinical work. | | | | | | | 3.59 | 0.73 |
| 1 | It is a very demanding and very difficult learning activity. | | | | | | | 3.59 | 0.83 |
| 19 | Created a good English learning environment. | | | | | | | 3.53 | 0.79 |
| 18 | For English courses, it is very appropriate. | | | | | | | 3.51 | 0.76 |
| 17 | Gave me a lot of motivation to improve my English. | | | | | | | 3.37 | 0.72 |
| 10 | I have benefited a lot from the writing feedback to my classmates. | | | | | | | 3.29 | 0.73 |
| 7 | The feedback from the students is very helpful for my writing improvement. | | | | | | | 3.24 | 0.74 |
| 2 | I really like to receive writing feedback and opinions from my classmates. | | | | | | | 3.06 | 0.59 |

These statements demonstrated not only the learners' insights about the online course, but they also revealed that learners of this generation have immense connectedness with the Internet. They can easily adapt to the use of technology as well as to online learning environments [24,26]. However, the students' responses implied another issue that worth considering. There is a discrepancy between the learning environment and the clinical settings. The nursing students could search unknown vocabularies online, and there were no time limitations during the writing, which is clearly inconsistent with the real-world situation of the workplace [6]. The physicians and/or nursing staff cannot look up a dictionary while writing a medical record or take a long time to write a medical record. The NNWC may improve ESP writing skills; however, there still exists a gap between the real-world situation [5].

The second question asked the learners for their perceptions about the peer review activity. Overall, 40% of the students replied that they were not sure whether their peers gave them the right feedback, nor could they (35%) identify others' mistakes. Although we had given proper PR training prior to the commencement of the writing tasks [22], the learners' lack of linguistic knowledge still hindered the quality of their feedback [13]. On the other hand, the third question asked learners about the TAs' feedback. Around 60% of the nursing students indicated that the most helpful aspect of the feedback from the TAs

and teacher was that they could indicate errors precisely and clearly. This was consistent with our findings in the CSQ and was like the findings of previous studies [7,13].

The fourth question was about the learners' perceptions regarding the eight writing topics. According to the responses, 33% of the learners perceived the fourth topic—colon cancer—to be the simplest. Since colon cancer has become one of the most prevalent cancers in Taiwan, most people are familiar with its symptoms and causes. This topic is quite close to the learners' daily lives, and therefore, the terms used seemed easy to learn. The topic's shorter amount of content also caused the learners to see it as the easiest one. In contrast, 33% of the students believed the second topic—heart attack—was the most difficult one. The patient discussed for that topic suffered from a non-ST-elevation myocardial infarction, had complex clinical symptoms, and underwent various medical treatments. Many medical terminologies had to be used and checked, and the caring plan was rather complicated compared with those for the other topics.

The fifth question asked the learners about their learning strategy when facing difficulties in doing ESP/ENP tasks. A total of 23% of the learners believed they should first enhance their general English competence [5], and 18% suggested utilizing Internet resources [32]. These two strategies correspond to the previous results indicating that EGP is crucial to the nursing students for ESP/ENP writing and that online writing is a preferred way of learning for them [8]. Next, when asked about their "needs" in terms of improving their writing skills, the learners perceived that grammar (46%) and vocabulary (46%) abilities were the most important aspects to be mastered. This was consistent with our previous findings that nursing students in such contexts lack confidence in their grammar and vocabulary competence when doing writing [7]. For the sixth question, in which we asked the students about "how" to ameliorate their writing skills, 54% of the learners answered that they should memorize more English words. The nursing students seemed to agree that sufficient vocabulary, including medical terms rather than grammatical knowledge, was critical for writing a nursing note [11].

Finally, the TPACK framework is worth considering when designing an ESP/ENP course, such as the current NNWC. Technologically, the online writing platform is recognized by the nursing students and echoes the previous literature [26,34]. Pedagogically, the process writing approach is a commonly used teaching method in writing research [23,38]. Contently, the EGP and ESP/ENP is the knowledge to be taught to the nursing students [1,6]. From the technologically pedagogical perspective, the process writing approach conducted in an online writing platform is an effective strategy and has been widely discussed in the literature [26,38]. From the technologically content aspect, teaching and learning ESP/ENP/EGP on the online writing platform was welcomed by the students and increased their writing performance significantly [26,34]. As to the pedagogically content knowledge integration, the process writing approach applied in ESP/ENP/EGP writing instruction has become a norm in contemporary language studies [13,23]. Comprehensively, integration of the technologically pedagogical content knowledge implemented in this study revealed the feasibility and effectiveness of this model and corresponds to the previous research as well [7,36,37].

## 5. Limitations

Five limitations to the methodological design of this study are acknowledged, including the following. Firstly, the study did not include a comparison group due to ethical concerns and the complex components of the nursing-note teaching. The effectiveness and efficiency of the course thus cannot be further tested through experimental manipulation. Secondly, the convenience sampling strategy can still pose potential hazards in terms of ethical issues and internal validity although the researchers adopted some strategies for avoiding such issues as collecting the data through research assistants located outside the classrooms after giving out the module scores, etc. Thirdly, that most nursing students were females in the researchers' teaching context is a bias regarding gender imbalance, which is another limitation for the research credibility. Fourthly, the overall sample size was

rather small such that the findings may not be suitable for generalization. Lastly, the online learning environment is not comparable with the clinical context, as the time constraints are different. Nursing students need to achieve a higher level of ESP/ENP proficiency before applying their note-taking skills to the healthcare industry.

## 6. Conclusions and Teaching Implications

This study demonstrates a comprehensive application of nursing note writing instruction based on the TPACK framework with an online writing platform as technological knowledge, process writing approach as pedagogical knowledge, and EGP/ESP/ENP as content knowledge. The results suggest an effective model that helped nursing students improve their writing performance in every dimension. The participants had significant improvement in writing during the period between pre-test and mid-test, and they gained more progress in ESP/ENP format than in the EGP format. On the other hand, the nursing students felt moderate satisfaction regarding the course design due to the demanding tasks and peer review activities although they recognized the value of the integration of technology into the syllabus. They also preferred receiving direct feedback and comments from their teacher regarding linguistic forms and content. As to the level of difficulties, "colon cancer" was mentioned to be the simplest, while "heart attack" was the hardest. In answering how to improve their writing competence, most responded that vocabulary and grammar, as well as general English competence, were crucial skills to be enhanced.

The purpose of this NWCC is to incorporate as many diverse and interactive learning opportunities as possible in the teaching of large classes. When designing the syllabus, language teachers are advised to accommodate various dimensions in addition to the traditionally cognitive-oriented focus, which was mainly on performance. Dimensions such as background knowledge stimulation, affective factors recognition, environmental creation, positive social interactions, etc., should be considered, too. The goal of our study is to develop a cognitive process in our students, that is, a norm that enables them to communicate with others through the texts under a common framework perceived by the public. By means of the comprehensive NWCC, students may construct their cognitive and learning progress through a diversified yet smooth and natural path.

In addition, an advantage of making use of nursing note-taking as a writing tool is to practice writing skills in real work situations. Nursing students are learning the English language differently from in the past. Some Internet tools, such as Google translation, have changed the importance and value of purely learning a foreign language (i.e., EGP). To motivate the nursing students, identifying the need of learning a foreign language in the clinical context is critical. The development of ESP/ENP capability embraces two abilities, which are English linguistic knowledge (EGP) and domain knowledge (nurse profession). The gaps between the EGP and the nurse profession can be bridged via the implementation of NWCC course.

Furthermore, it might still be true that most Taiwanese nurses "code switch" when they write nursing notes. Code switch refers to the situation in which language users mix two languages when writing. Clinically, Taiwanese nurses often use medical terminologies in English since the physicians are writing prescriptions and orders in English. English medical terminologies have become the commonly used communication tool among doctors and nurses. However, nurses are expressing general ideas in Chinese because they lack EGP competencies. This phenomenon shows there is room for improvement in English proficiency in the overall nursing profession, and it reveals the reason why this research and its derivative curriculum are important.

Future studies including different academic subjects and/or specific model dimensions based on nursing note teaching are recommended. Inclusions of more male participants and the design of a control group in the future may add to the rigor of the study, too.

**Author Contributions:** Conceptualization, S.-M.S.; methodology, H.-C.T.; data validation, S.-M.S.; formal analysis, H.-C.T.; investigation, S.-M.S.; resources, Y.-H.T.; writing, H.-C.T.; review and editing, S.-M.S.; proofread, H.-C.T.; supervision, H.-C.T.; project administration, H.-C.T.; funding acquisition, Y.-H.T. All authors have read and agreed to the published version of the manuscript.

**Funding:** This paper, as part of two research projects, was funded by the Ministry of Science and Technology (MOST 107-2410-H-255-001-) and Chang Gung Memorial Hospital (CMRPF6H0061). The funding sources only provided the financial assistance without any involvement during the research process.

**Institutional Review Board Statement:** The study was conducted according to the guidelines of the Declaration of Helsinki and was approved by the Institutional Review Board of Chang Gung Medical Foundation (protocol code: 201800874B0 and date of approval: 20180626).

**Informed Consent Statement:** Informed consent was obtained from all subjects involved in the study.

**Data Availability Statement:** The data presented in this study are available on reasonable request from the corresponding author.

**Acknowledgments:** We, as a team, give special thanks to the research assistant Liu Jia-xin who offered her time and effort to assist in the completion of this project. All participants who volunteered for this study are also noted.

**Conflicts of Interest:** The authors declare no conflict of interest. The funder played no role in the design of the study; in the collection, analyses, or interpretation of data; in the writing of the manuscript; or in the decision to publish the results.

## Appendix A. Teaching Schedule of Nursing Note Writing

**Table A1.** Teaching schedule of the NNWC.

| Week | Teaching Progress | Writing Task | Note |
|---|---|---|---|
| 1 | Course introduction | Nursing note-writing pre-test (prostate enlargement) | |
| 2 | Introductory session to nursing note writing (grading criteria, essay presentation, common mistakes, etc.) and other peer feedback form description and training. | Peer feedback examples and trials | • 4 people in a group, choose a group leader as the contact window. |
| 3 4 5 | Topic 1: Gastroenterology | Cirrhosis of the liver PR Feedback (first draft) PR Feedback (draft amendments) | • After class, students engage in peer feedback and collaborative learning online. |
| 6 7 8 | Topic 2: Cardiovascular | Heart disease PR Feedback (first draft) PR Feedback (draft amendments) | • The writing process is recorded online. • Participation in the process and composition will be counted as 10% of the semester grade. |
| 9 10 | Mid-term exam  Video: spring in the emergency room | Nursing record-writing mid-term test (diabetes) | • Essays are submitted to peers and teachers for review online. |
| 11 12 13 | Topic 3: Thoracic | Tuberculosis (TB) PR Feedback (first draft) PR Feedback (draft amendments) | • Teachers give group or individual feedback online. • Teachers give feedback online as appropriate. |
| 14 15 | Topic 4: Orthopedics | Colon cancer PR Feedback | |
| 16 17 | Topic 5: Breast Surgery Department | Breast cancer PR Feedback | |
| 18 | Final exam | Nursing note-writing post-test (stroke) | Writing course feedback questionnaire |

**Appendix B. EGP Format of Nursing Note = Writing Tasks**

Class:_____________      Seat               number:_____________      Name:_____________
Date:___________(mmdd)
Foreword: first draft:_____; draft amendment:_____; completed draft:_____ (please check V where appropriate.)
Note: The following is a nursing note written in Chinese. Please rewrite it into English using full sentences and appropriate terminologies.

---

這位 52 歲的男性病患，三個月前有非 ST 段上升心肌梗塞病史，被診斷有糖尿病及高血壓。病人呈現輕微的胸悶，但沒有胸痛，小便量正常，沒有下肢水腫。三個月前鉈掃描結果顯示，心尖部位有缺血、及心臟側面有梗塞。在我們急診室，心電圖剛開始呈現竇性心搏過速，後來轉變為心房纖維顫動、且心室反應快速，然而病人血液動力穩定，發現他有心跳不規律、伴有二級的收縮性心雜音、以及心肌酵素上升。病人住進我們病房，作進一步的評估和治療，以排除為:非 ST 段上升心肌梗塞合併急性肺水腫的可能。

---

(After completing the job, please indicate the number of words: __ 128___ words)
詳述:_The 52-year-old male patient was diagnosed with diabetes mellitus and hypertension. He had medical history of non-ST-elevation myocardial infarction found three months ago. The patient appeared mild chest tightness without chest pain. The amount of urine was at a normal level, and he didn't have lower leg edema. The thallium scan revealed ischemia in the apical segment and infarction in the lateral segment. At our ER, the patient's electrocardiogram revealed sinus tachycardia first, and then changed to atrial fibrillation with rapid ventricular response. However, the case was in a stable hemodynamic condition. He was found irregular heartbeat with a systolic murmur, grade II/VI, and rising myocardial enzyme. The patient was admitted to our ward for further evaluation and treatment to rule out non-ST-elevation MI with acute pulmonary edema.

| Content | Organizational Structure | Grammar | Words, Spelling | Example (Format, Punctuation, Case) | Total Score |
|---|---|---|---|---|---|
| | | | | | |

**Appendix C. ESP/ENP Version of Nursing Note Writing Tasks**

Class:_____________      Seat               number:_____________      Name:_____________
Date:___________(mmdd)
Foreword: first draft:_____; draft amendment:_____; completed draft:_____ (please check V where appropriate.)
Note: The following is a nursing note written in Chinese. Please rewrite it into English using full sentences and appropriate terminologies.

---

這位 52 歲的男性病患，三個月前有非 ST 段上升心肌梗塞病史，被診斷有糖尿病及高血壓。病人呈現輕微的胸悶，但沒有胸痛，小便量正常，沒有下肢水腫。三個月前鉈掃描結果顯示，心尖部位有缺血、及心臟側面有梗塞。在我們急診室，心電圖剛開始呈現竇性心搏過速，後來轉變為心房纖維顫動、且心室反應快速，然而病人血液動力穩定，發現他有心跳不規律、伴有二級的收縮性心雜音、以及心肌酵素上升。病人住進我們病房，作進一步的評估和治療，以排除為:非 ST 段上升心肌梗塞合併急性肺水腫的可能。

---

(After completing the task, please indicate the number of words: __ 82___ words)

簡述: The 52 y/o male P't had NSTEMI & D.M & H/T Hx for 3 months. Has mild chest tightness w/o chest pain, normal urine amount & no lower leg edema. Thallium scan: ischemia in apical seg & infarction in lateral seg. At ER, ECG: ST & then became Af with RVR. In stable hemodynamic condition. Irregular heartbeat with systolic murmur gr 2/6, & cardiac enzymes (CK-MB) elevation. P't admitted for further Tx to R/O NSTEMI with APE.

| Content | Words, Spelling (Including the Use of Technical Terms) | Punctuation and Capital-ization | Nursing Record Special Grammar | Other General Grammar | Total Score |
|---|---|---|---|---|---|
| | | | | | |

## Appendix D. Peer Review Form for ESP/ENP Writing

**Table A2.** Nursing note writing peer review form.

Topic:_______________ Writer's name:__________ Seat number:________
Name of peer reviewer:________ Seat number:________ Review Date:_______(2018)
Overall score:________ (0 to 5 point)

| No. | Review Items | Initial Draft | | |
|---|---|---|---|---|
| | | Yes | No | Suggestions |
| | Content: | | | |
| (1) | The writing completely expressed the care situation. | | | |
| (2) | The content is organized smoothly. | | | |
| (3) | The description is clear and easy to understand. | | | |
| | Words and spelling: | | | |
| (4) | Vocabulary was used correctly. | | | |
| (5) | Words spelled correctly. | | | |
| (6) | Technical terms were used correctly. | | | |
| | Punctuation and capital words: | | | |
| (7) | Abbreviations were correctly used. | | | |
| (8) | Proper nouns and names were capitalized. | | | |
| (9) | Commas correctly used. | | | |
| (10) | Periods used at the end of sentences. | | | |
| (11) | Avoid using Chinese punctuation. | | | |
| (12) | Capitalize the first word in a sentence. | | | |
| | Nursing Note Grammar: The writer is able to . . . | | | |
| (13) | The writer is able to omit the subject in a sentence correctly. | | | |
| (14) | Omit the object in a sentence correctly. | | | |
| (15) | Omit the subject and the verb correctly. | | | |
| (16) | Omit the article correctly (e.g., *a* or *the*). | | | |
| (17) | Omit the *be* verb correctly. | | | |
| (18) | Omit the passive *be* verb correctly. | | | |
| (19) | Use the imperative correctly. | | | |
| | Other general grammar: | | | |
| (20) | The writer uses phrases correctly. | | | |

**Table A2.** *Cont*.

| | |
|---|---|
| (21) | The writer uses tense correctly. |
| (22) | The writer adds "*s*" to the third-person singular verbs. |
| (23) | Use pronouns (nominative/possessive/qualifier) correctly. |
| (24) | Adding "s" to plural countable nouns. |
| (25) | Use prepositions correctly. |
| (26) | Use adverbs correctly. |
| (27) | Use adjectives correctly. |
| (28) | Use auxiliary verbs correctly. |
| (29) | Any other unlisted errors. |

The most appreciated student of this nursing note-writing task and reasons:

Suggestions or comments by the teacher or TA:

Note: 1. Reviewers should try their best to help their peers find errors as much as possible. 2. Please check "Yes" if do not find any mistakes in each item. 3. Check "No" if you find any mistake in each item. 4. Before each scoring item "label", students can make good use of filling in the essay in order to facilitate identification. 5. In the "suggestions" column, please provide suggestions as far as possible according to the error. 6. Give an appropriate overall score according to the GEPT criteria. 7. Finally, don't forget to give your classmates some applause and encouragement; write some good points. 8. Please do not fill in the "suggestions or comments by the teacher or TA", teachers or TA complement deficiencies.

## Appendix E. Learners' Perceptions about the Writing Course

**Table A3.** The Course Satisfaction Questionnaire.

1. For online writing or paper–pencil writing, you prefer ☐ Online ☐ Paper–pencil
   Reasons: _________________________________________

2. What do you think of the ratings and design of peer feedback?
   (1) The difficulty of peer feedback: _________________________

   (2) How to overcome this difficulty: _________________________

3. For the feedback of "writing teacher and online TAs", you think:
   (1) Helpful: ______________________________________________

   (2) Difficult: ____________________________________________

   (3) How to overcome this difficulty: _______________________

4. Which nursing note do you think is the easiest? Which one is the hardest? Why?
   (1) The easiest: _____________________
       Reasons: _____________________

   (2) The hardest: _____________________
       Reasons: _____________________

5. In your nursing note writing, what do you think you can improve on?
   ___________________________________________________

   And how? ______________________________________________

6. Finally, do you have any relevant suggestions for this semester's nursing note-writing course?
   ___________________________________________________

## Appendix F. Learners' EGP Test Scores (School Quiz, Midterm, and Final Tests)

**Table A4.** The nursing students' EGP performance during the research period.

| Seat No. | Quiz | Midterm | Final Test | Seat No. | Quiz | Midterm | Final Test |
|----------|------|---------|------------|----------|------|---------|------------|
| 1 | 100 | 85 | 86 | 27 | 99 | 94 | 82 |
| 2 | 100 | 87 | 86 | 28 | 100 | 66 | 81 |
| 3 | 96 | 69 | 85 | 29 | 100 | 84 | 84 |
| 4 | 100 | 53 | 56 | 30 | 96 | 80 | 83 |
| 5 | 100 | 89 | 77 | 31 | 100 | 93 | 92 |
| 6 | 100 | 92 | 91 | 32 | 94 | 78 | 69 |
| 7 | 81 | 83 | 86 | 33 | 100 | 93 | 96 |
| 8 | 97 | 79 | 86 | 34 | 93 | 60 | 70 |
| 9 | 98 | 89 | 72 | 35 | 98 | 71 | 83 |
| 10 | 94 | 71 | 45 | 36 | 100 | 85 | 89 |
| 11 | 99 | 77 | 84 | 37 | 98 | 63 | 64 |
| 12 | 99 | 95 | 92 | 38 | 98 | 90 | 85 |
| 13 | 100 | 92 | 94 | 39 | 99 | 75 | 70 |
| 14 | 100 | 93 | 94 | 40 | 99 | 88 | 86 |
| 15 | 80 | 64 | 64 | 41 | 94 | 77 | 74 |
| 16 | 100 | 74 | 67 | 42 | 99 | 93 | 100 |
| 18 | 97 | 88 | 91 | 43 | 99 | 95 | 94 |
| 19 | 100 | 66 | 78 | 44 | 99 | 87 | 80 |
| 20 | 99 | 79 | 86 | 45 | 100 | 83 | 67 |
| 21 | 96 | 88 | 87 | 46 | 100 | 93 | 94 |
| 22 | 100 | 84 | 89 | 47 | 91 | 92 | 96 |
| 23 | 94 | 89 | 96 | 48 | 99 | 87 | 92 |
| 24 | 99 | 86 | 95 | 49 | 100 | 97 | 99 |
| 25 | 100 | 90 | 79 | 50 | 99 | 90 | 92 |
| 26 | 100 | 83 | 87 | Mean | 97.13 | 81.83 | 81.97 |

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
