# Peer review of "An ESP Approach to Teaching Nursing Note Writing to University Nursing Students"

_education, doi:10.3390/educsci12030223_

Round 1

Reviewer 1 Report

The issue addressed is quite relevant for the vocational training of nurses; the study also contributes to the discussion of developing writing skills in ESP.

The manuscript is clear and well-structured, the argument presented is scientifically sound.

Nevertheless, there are several comments which could help provide necessary details and thus improve the overall quality of the manuscript:

  1. In Figure 1 the sample for the experiment is described. The sample included 49 students in total, with the number of male and female students 27 and 2, correspondingly. In Section 2.2 the number of male and female students is different (47 + 2). It seems to be an error in Fig. 1.
  2. Figure 1 presents an online writing platform as a teaching strategy. It seems to be more appropriate to designate a platform as a tool, which can be used to implement the strategies mentioned (peer review, teacher feedback, etc.).
  3. In Section 2.2 the author(s) provide information about the English learning experiences that the students in the experimental sample have. Probably, it would be informative to indicate the estimated level of their English language proficiency according to CEFR?
  4. Figure 2 presents the screenshot of the online writing platform page. Could the author(s) provide an English translation of the key points to make the figure more understandable for the readers who do not speak Mandarin Chinese?
  5. Figure 3 (p.4) shows the algorithm of the multiple drafts teaching strategy. The full procedure is repeated 5 times. Is it related to the number of topics? In this case, it is worth mentioning it explicitly.
  6. Section 2.4 describes five writing tasks used in the experiment. It would be informative to present at least one example of the tasks mentioned to provide a better understanding for readers.
  7. Table 2 provides the results of the tests. The parameters measured include six dimensions (content, structure, grammar, spelling, mechanism, and holistic). Could the author(s) specify in the text, which aspects are measured as mechanics? Does mechanics include spelling?
  8. The EGP test results (Table 2) show that the mean scores for Spelling and Structure decreased in the post-test compared to the mid-test. A brief comment could be provided explaining the possible causes for this decrease.

Author Response

Thank you so much for the efficient review process and valuable feedback from the reviewers. We are submitting a revised manuscript entitled “An ESP Approach to Teaching Nursing Note Writing to University Nursing Students” to the Journal of Education Science for the consideration of publish. To respond the reviewer’s comments and suggestions, we list the questions point by point and the revisions we have made on the manuscript as follows.

Comment: The issue addressed is quite relevant for the vocational training of nurses; the study also contributes to the discussion of developing writing skills in ESP.
The manuscript is clear and well-structured, the argument presented is scientifically sound.
Nevertheless, there are several comments which could help provide necessary details and thus improve the overall quality of the manuscript:

Re: Dear reviewer, thank you very much for the comment. We are trying our best to improve the quality of this manuscript according to your feedback and expertise.

Q 1. In Figure 1 the sample for the experiment is described. The sample included 49 students in total, with the number of male and female students 27 and 2, correspondingly. In Section 2.2 the number of male and female students is different (47 + 2). It seems to be an error in Fig. 1.

Re: Thanks a lot for the suggestion. The Figure 1 has been changed into Figure 2, and the figures of participants have been corrected. Please see revisions marked by highlighted fluorescence on Page 5, Line 211.

Q 2. Figure 1 presents an online writing platform as a teaching strategy. It seems to be more appropriate to designate a platform as a tool, which can be used to implement the strategies mentioned (peer review, teacher feedback, etc.).

Re: Thanks a lot for the suggestion. The Figure 1 has been changed into Figure 2, and the research procedures have been modified. We added another Figure 1 to demonstrate the research framework which clearly define the online writing platform as a technological knowledge in the TPACK model. Please see revisions marked by highlighted fluorescence on Page 5, Line 211.

Q 3. In Section 2.2 the author(s) provide information about the English learning experiences that the students in the experimental sample have. Probably, it would be informative to indicate the estimated level of their English language proficiency according to CEFR?

Re: Thanks a lot for the suggestion. The participants’ EGP proficiency has been added into the text. Please see revisions marked by highlighted fluorescence on Page 6, Line 233-242.

Q 4. Figure 2 presents the screenshot of the online writing platform page. Could the author(s) provide an English translation of the key points to make the figure more understandable for the readers who do not speak Mandarin Chinese?

Re: Thanks a lot for the suggestion. The Figure 2 has been changed into Figure 3, and we found the English version of this webpage. Please see revisions marked by highlighted fluorescence on Page 7, Line 266-267.

Q 5. Figure 3 (p.4) shows the algorithm of the multiple drafts teaching strategy. The full procedure is repeated 5 times. Is it related to the number of topics? In this case, it is worth mentioning it explicitly.

Re: Thanks a lot for the suggestion. The rationale of 5 times repeated procedures has been stated in the text. Please see revisions marked by highlighted fluorescence on Page 8, Line 293-298.

Q 6. Section 2.4 describes five writing tasks used in the experiment. It would be informative to present at least one example of the tasks mentioned to provide a better understanding for readers.

Re: Thanks a lot for the suggestion. Two samples of the writing tasks, including an EGP (appendix 2) and an ESP/ENP writing (appendix 3), have been added to the appendixes. Please see revisions marked by highlighted fluorescence on Page 21 & 22.

Q 7 Table 2 provides the results of the tests. The parameters measured include six dimensions (content, structure, grammar, spelling, mechanism, and holistic). Could the author(s) specify in the text, which aspects are measured as mechanics? Does mechanics include spelling?

Re: Thanks a lot for the suggestion. The scope of the mechanics dimension has been added to the learners’ competences measurement as “whilst the mechanics dimension dealt with the punctuation, capital letters, transitional words, and discourse markers”. Please see revisions marked by highlighted fluorescence on Page 10, Line 393-394.

Q 8 The EGP test results (Table 2) show that the mean scores for Spelling and Structure decreased in the post-test compared to the mid-test. A brief comment could be provided explaining the possible causes for this decrease.

Re: Thanks a lot for the suggestion. The possible causes and comments have been added to the text. Please see revisions marked by highlighted fluorescence on Page 13, Line 502-516.

Should you have any further concerns, please feel free to contact us.

With Best Regards,

The Authors

Reviewer 2 Report

A brief summary
The study describes a nursing  note writing course (NNWC) by using five teaching tools, including the online writing platform、multiple revisions、peer review activities、direct and indirect teacher feedback in order to give university nursing students English for specific purposes (ESP) training. 

Broad comments
The pedagogical background of the study is absent. It is not clear if the procedure is based on any theoretical background.
In the discussion and conclusions section, the authors seem to provide a summary of their work and nothing further. I cannot find any major contribution from the discussion or conclusions except the online writing platform. What is the major contribution comparing with the traditional way? 
The authors, in this section, should link the conclusions derived from their findings with the results obtained by other researchers in similar works. Referencing, obviously, the authors of these works.

Specific comments

  • Research contribution is a bit poor.
  • literature review is totally missing.
  • The appendix 1、appendix 2、appendix 3、appendix 4 and appendix 5 are missing.

Author Response

Thank you so much for the efficient review process and valuable feedback from the reviewers. We are submitting a revised manuscript entitled “An ESP Approach to Teaching Nursing Note Writing to University Nursing Students” to the Journal of Education Science for the consideration of publish. To respond the reviewer’s comments and suggestions, we list the questions point by point and the revisions we have made on the manuscript as follows.

Q 1. The pedagogical background of the study is absent. It is not clear if the procedure is based on any theoretical background.

Re: Thanks a lot for the suggestion. The research design and its theoretical framework have been added into section 3.1 and Figure 1. This is an action research based on the framework of TPACK model. Please see revisions marked by highlighted fluorescence on Page 4, Lines 195-207.

Q 2. In the discussion and conclusions section, the authors seem to provide a summary of their work and nothing further. I cannot find any major contribution from the discussion or conclusions except the online writing platform. What is the major contribution comparing with the traditional way? The authors, in this section, should link the conclusions derived from their findings with the results obtained by other researchers in similar works. Referencing, obviously, the authors of these works.

Re: Thanks a lot for the suggestion. We’ve added the literature review part, theoretical framework, discussion, and referencing. Please refer to the revisions marked by highlighted fluorescence in the whole manuscript.

Q 3. Literature review is totally missing.

Thanks a lot for the suggestion. The 4 sections of literature review have been added from section 2.1 to 2.4. Please see revisions marked by highlighted fluorescence on Page 2-4.

Q 4. The appendix 1, appendix 2, appendix 3, appendix 4, and appendix 5 are missing.

Re: Thanks a lot for the suggestion. The 6 appendixes have been added to the end of the manuscript. Please see revisions marked by highlighted fluorescence on Page 20-26.

Should you have any further concerns, please feel free to contact us.

With Best Regards,

The Authors

Reviewer 3 Report

  1. Due to ESP capability embraces two abilities, English linguistic knowledge (i.e. EGP) and the domain knowledge (i.e. nurse profession). I’d like to know the participants’ EGP proficiency, hence, please add their English certification scores (e.g. TOFEL, TOEIC, GEPT, etc.) or the updated English test scores (e.g. school midterm or final tests) into “2.2. Participants” section.

  1. The manuscript missed all the Appendixes, please add those after the References section.

  1. I suggested that the content of the sub-section “Nursing note writing tasks” should be integrated into “2.3. Teaching interventions”.

  1. I suggested that integrating “2.4. Data collection” and “2.5. Data analysis” to become “2.4 Data collection and processing”.

  1. Formal professional English writing skills for nurses is certainly important because when an EFL nurse serves in international hospital or medical activities, it will be an important skill, however, after reading this paper, I just realized the ESP writing that the authors mentioned was the so-called note taking or note writing, please mention why taking note in English for nurses is important? For example, when a nurse taking a note, who will read the note? In Taiwan, if the note is for the nurse of next shift to read, it is better to record in Chinses language to avoid misunderstanding (except the terminology may remain in English), isn’t it? Thus, the explanation of “why a Taiwanese nurse need to take the so-called note in English” should be clarified.

  1. From your qualitative analysis of the 6 open-ended questions, I just discovered the participants could search unknown vocabularies on line and there were no time limitations during the writing, which is clearly inconsistent with the real-world situation of the workplace. In other words, have you ever seen a heal care worker (e.g. a physician) look up a dictionary while writing a medical record or take a long time to write a medical record. Hence, it is necessary to add this to your limitation. The NNWC may improve ESP writing skills; but there still exist a gap between the real-world situation.

“When I need a word to express my thought, I can look it up in the online dictionary immediately. Online writing can help me increase the amount of vocabulary effectively.” (ID. 7)

“There are no space and/or time restrictions to do online writing, and the tasks can be revised/ proofread/ marked efficiently.” (ID. 20)

  1. The authors should re-check the overall grammars, spellings, integrate and trim simple sentences to improve its readability.

Author Response

Thank you so much for the efficient review process and valuable feedback from the reviewers. We are submitting a revised manuscript entitled “An ESP Approach to Teaching Nursing Note Writing to University Nursing Students” to the Journal of Education Science for the consideration of publication. To respond the reviewer’s comments and suggestions, we list the questions point by point and the revisions we have made on the manuscript as follows.

Q 1. Due to ESP capability embraces two abilities, English linguistic knowledge (i.e. EGP) and the domain knowledge (i.e. nurse profession). I’d like to know the participants’ EGP proficiency, hence, please add their English certification scores (e.g. TOFEL, TOEIC, GEPT, etc.) or the updated English test scores (e.g. school midterm or final tests) into “2.2. Participants” section.

Re: Thanks a lot for the suggestion. The participants’ EGP proficiency has been added into the text. Please see revisions marked by highlighted fluorescence on Page 6, Line 234-243.
In addition, as the nursing students’ English certification scores are not available, we provide their updated English test scores in appendix 6 on Page 26.

Q 2. The manuscript missed all the Appendixes, please add those after the References section.

Re: Thanks a lot for the suggestion. The 6 appendixes have been added to the end of the manuscript. Please see revisions marked by highlighted fluorescence on Page 20-26.

Q 3. I suggested that the content of the sub-section “Nursing note writing tasks” should be integrated into “2.3. Teaching interventions”.

Re: Thanks a lot for the suggestion. The abovementioned section has been moved to integrate with the section of “teaching intervention”. Please see revisions marked by highlighted fluorescence on Page 9, Lines 332-370.

Q 4. I suggested that integrating “2.4. Data collection” and “2.5. Data analyses” to become “2.4 Data collection and processing”.

Re: Thanks a lot for the suggestion. The abovementioned two sections have been combined together to become the section of “Data collection and processing”. Please see revisions marked by highlighted fluorescence on Page 9-10, Lines 371-421.

Q 5. Formal professional English writing skills for nurses is certainly important because when an EFL nurse serves in international hospital or medical activities, it will be an important skill, however, after reading this paper, I just realized the ESP writing that the authors mentioned was the so-called note taking or note writing, please mention why taking note in English for nurses is important? For example, when a nurse taking a note, who will read the note? In Taiwan, if the note is for the nurse of next shift to read, it is better to record in Chinses language to avoid misunderstanding (except the terminology may remain in English), isn’t it? Thus, the explanation of “why a Taiwanese nurse need to take the so-called note in English” should be clarified.

Re: Thanks a lot for the suggestion. As we removed the section of literature review in the previous version, the manuscript did not mention the needs analysis of the ESP/ENP context. We now add a whole part of literature review which explains the rationale of conducting this research. Please see revisions marked by highlighted fluorescence on Page 2, Lines 58-90.
In addition, two paragraphs regarding teaching implications of this study are added to the section of “conclusion and teaching implications”. Please see revisions marked by highlighted fluorescence on Page 16-17, Lines 638-655.

Q 6. From your qualitative analysis of the 6 open-ended questions, I just discovered the participants could search unknown vocabularies on line and there were no time limitations during the writing, which is clearly inconsistent with the real-world situation of the workplace. In other words, have you ever seen a heal care worker (e.g. a physician) look up a dictionary while writing a medical record or take a long time to write a medical record. Hence, it is necessary to add this to your limitation. The NNWC may improve ESP writing skills; but there still exist a gap between the real-world situation.

“When I need a word to express my thought, I can look it up in the online dictionary immediately. Online writing can help me increase the amount of vocabulary effectively.” (ID. 7)

“There are no space and/or time restrictions to do online writing, and the tasks can be revised/ proofread/ marked efficiently.” (ID. 20)

Re: Thanks a lot for the suggestion. As we removed the section of literature review in the previous version, the manuscript did not mention the needs analysis of the ESP/ENP context. We now add a whole part of literature review which explains the rationale of conducting this research. Please see revisions marked by highlighted fluorescence on Page 2.

Q 7 The authors should re-check the overall grammars, spellings, integrate and trim simple sentences to improve its readability.

Re: Thanks a lot for the suggestion. We will apply for the journal’s English editing services to further improve the quality of this manuscript.

Should you have any further concerns, please feel free to contact us.

With Best Regards,

The Authors

Round 2

Reviewer 3 Report

This paper can be accepted after revision. Before published, please re-check the spelling and grammar.